# Incorporating Bilingual Dictionaries for Low Resource Semi-Supervised Neural Machine Translation

**Sreyashi Nag** [*], **Mihir Kale** [*], **Varun Lakshminarasimhan** [*]**& Swapnil Singhavi**
Language Technologies Institute
Carnegie Mellon University
Pittsburgh, PA 15213, USA
{sreyashn,mihirsak,vbl,ssinghav}@cs.cmu.edu

## Abstract

We explore ways of incorporating bilingual dictionaries to enable semi-supervised neural machine translation. Conventional back-translation methods have shown success in leveraging target side monolingual data. However, since the quality of back-translation models is tied to the size of the available parallel corpora, this could adversely impact the synthetically generated sentences in a low resource setting. We propose a simple data augmentation technique to address both this shortcoming. We incorporate widely available bilingual dictionaries that yield word-by-word translations to generate synthetic sentences. This automatically expands the vocabulary of the model while maintaining high quality content. Our method shows an appreciable improvement in performance over strong baselines.

## 1 Introduction

Neural Machine Translation (NMT) methods require large amounts of parallel data to perform well. This poses a significant challenge in low-resource and out-of-domain scenarios where the amount of parallel data is usually limited. A proven way to mitigate this issue has been by leveraging the vast amounts of monolingual data in conjunction with parallel data to improve performance. Prior work in the field has explored several methods to achieve this. One of the most successful approaches has been Back-Translation (BT) Sennrich et al. (2015), that generates artificial parallel data from target monolingual corpora by training a translation model in the reverse direction. Another approach (COPY) proposed by Currey et al. (2017) directly copies target monolingual data to the source, focused on capturing entities that do not change across languages.

The methods mentioned above suffer from a couple of limitations. The quality of the generated source translations in the BT model are dependent on the amount of parallel data. Furthermore, the vocabulary available to the model is also limited to that of the parallel data, which increases the probability of out-of-vocabulary words. The COPY model, on the other hand, adds vocabulary, albeit only on the target side. In this paper, we propose a simple yet effective data augmentation technique that utilizes bilingual dictionaries that expands vocabulary on both source and target sides, thus significantly reducing the probability of out-of-vocabulary words. Our method also ensures that correlations between the source and target languages are modelled in the monolingual data. In particular, our contributions are as follows:

- We propose the Word-on-Word (WoW) data augmentation method, that outperforms previous data augmentation methods in a low-resource setting.

- We show that our method benefits from both in-domain as well as out-of-domain monolingual data and shows encouraging results for domain-adaptation.

- Finally, we also apply our method over other augmentation techniques and show its effectiveness in enhancing performance.

---

[*]Equal contribution

## 2 RELATED WORK

Back-translation Sennrich et al. (2015) has emerged as a popular way of using monolingual data on the target side. Burlot & Yvon (2018) show that the quality of the reverse model directly impacts translation quality - augmenting data generated from a weak backtranslation model leads to only small improvements. Our method directly addresses this issue by utilizing high-quality bilingual dictionaries. Zhang & Zong (2016b) consider using data on the source side in a self-training setup. Other ways of data augmentation include copying target data on the source side Currey et al. (2017). Sennrich et al. (2015) use target side monolingual data by using null sentences on the source side, effectively performing language modelling as an auxiliary task. Another way of incorporating monolingual data is via hidden states from pre-trained language models, as done by Gulcehre et al. (2015).

In terms of incorporating bilingual dictionaries into NMT, Zhang & Zong (2016a) use them for data augmentation. However, they focus mainly on rare words, and unlike our approach, their method has a dependency on statistical phrase-based translation models. Arthur et al. (2016) use translation lexicons, but their objective is to use them for influencing the probability distribution of the decoder. Word-by-word translation has also been used for unsupervised translation Lample et al. (2017), while our goal is to utilize it in the semi-supervised setup.

## 3 EXPERIMENTAL SETUP

We use the TED Talks corpus Qi et al. (2018) with the provided train, dev and test splits. Specifically we consider German-English (*de-en*) and Spanish-English (*es-en*) translation tasks. We use subsets from the given train split to simulate low-resource settings. We use freely available bilingual dictionaries provided by Facebook[1]. For our experiments, we employ a 1-layer 256 dimensional encoder-decoder model with attention Bahdanau et al. (2014) with a beam width of 5 for decoding. Training uses a batch size of 32 and the Adam optimizer Kinga & Adam (2015) with an initial learning rate of 0.001, with cosine annealing. Models are trained for 50 epochs, while model selection is performed according to performance on the validation set using BLEU Papineni et al. (2002).

| Target English | The work of a transportation commissioner isn't just about stop signs and traffic signals . |
|---|---|
| Ground Truth | El trabajo de una Comisaria de Transporte o es solo sobre seales de  pare  y semforos . |
| Copied Target (COPY) | The work of a transportation commissioner isn't just about stop signs and traffic signals . |
| Back Translation (BT) | el trabajo de una evolucin funcional no est hablando con los altibajos y aman a las seales . |
| Bilingual Dictionary | trabajo del una transportacin comisionado just acerca pare letreros trfico seales |

Table 1: Comparison of various data augmentation techniques

## 4 METHOD

Our approach utilizes bilingual dictionaries to obtain word-on-word translations (WoW) for target side monolingual data. Given a sentence in the target language (in our case, English), we apply the dictionary on each word to obtain corresponding translations in the source language. We then augment our parallel corpus with this synthetically created data on the source side and the ground truth monolingual data on the target side. We then train our model on this augmented dataset to achieve our final translations. Figure 1 shows the popular approaches of data augmentation using monolingual corpora on the target side and how they compare with our proposed approach.

The main benefit of WoW over back-translation (BT) is in the quality of the generated synthetic

---

[1]https://github.com/facebookresearch/MUSEground-truth-bilingual-dictionaries

data. BT in very low-resource settings results in a poor model, which in turn generates poor quality synthetic sentences. WoW on the other hand, relies on strong bilingual dictionaries. Hence, even if the sentence structure due to the word-on-word translation is poor, it at least ensures that the words in the synthetic sentences are accurate. With the right words on the source side and an approximately correct word ordering, the model has a rough sketch of the semantics of the sentence.

Another clear benefit of this method is that it allows the model to expand its vocabulary. For instance, starting with 10k parallel pairs, adding 10K WoW pairs helps increase vocabulary coverage on the development set from 65% to 92% for *es-en* on the target side. Note that the vocabulary expansion effect is both on the target as well as source side. On the source side, the model is exposed to a lot fewer *unks*, and the coverage increases from 60% to 88%. Theoretically, COPY also expands vocabulary on the target side. However, it does so in a way that is independent of the source sentence. With WoW on the other hand, the model can make direct correlations between new target words and the source words. Note that for every pair in the augmented dataset, the target sentence is always a high quality, real world sentence. The synthetic data is only added to the source side, and the quality of the supervised labels for the decoder remain untarnished.

## 5 RESULTS AND ANALYSIS

### 5.1 TRANSLATION PERFORMANCE

We compare WoW with 3 baselines :
- **Parallel**: Only 10k parallel data.
- **BT**: The parallel corpus is augmented with 10K back-translated pairs.
- **COPY**: The parallel corpus is augmented with 10K copied target data.

As shown in table 2, BT outperforms the parallel-only baseline, showing that even weak synthetic sentences can improve BLEU scores. However, BT itself is beaten by the rather simple COPY method. The low performance of BT compared to COPY can be attributed to the poor quality source sentences that it generates.

WoW outperforms all baselines, including COPY, for both language pairs. WoW beats the best data augmentation baseline (COPY) by 0.85 points for *es-en* and 0.79 points for *de-en*. Gains over parallel-only data are 2.5 for *de-en* and 2.8 for *es-en*. For comparison, we also report BLEU scores for 20k parallel data, which gives us an upper bound on what data augmentation techniques can achieve.

| experiment | de-en | es-en |
|---|---|---|
| parallel 10k | 9.46 | 9.51 |
| + 10k BT | 10.86 | 11.47 |
| + 10k COPY | 11.24 | 11.49 |
| + 10k WoW | 12.03 | 12.34 |
| + 10k parallel | 13.01 | 13.46 |

Table 2: Comparison with Baselines

| experiment | de-en | es-en |
|---|---|---|
| parallel 10k | 9.46 | 9.51 |
| + 10k WoW | 12.03 | 12.34 |
| + 20k WoW | 12.66 | 12.96 |
| + 40k WoW | 13.31 | 13.79 |
| + 10k parallel | 13.01 | 13.46 |

Table 3: Increasing Monolingual Data

### 5.2 EFFECT OF SIZE OF MONOLINGUAL DATA

We perform further experiments, increasing the size of the monolingual data used for augmentation. Three settings are considered: 1:1 (10k synthetic), 1:2 (20k synthetic) and 1:4 (40k synthetic) ratios for the parallel and synthetic data respectively. For both language pairs, we see substantial improvements - upto 3.8 BLEU points for *de-en* and 4.3 points for *es-en* - with the increase in monolingual data. This can be seen in table 3. We observe that adding 40k synthetic sentences brings about more benefit than adding 10k high quality parallel sentences.

| experiment | de-en | es-en |
|---|---|---|
| parallel 10k | 9.46 | 9.51 |
| + 10k WoW | 11.59 | 12.27 |
| + 20k WoW | 12.36 | 12.73 |
| + 40k WoW | 13.0 | 12.91 |
| + 10k parallel | 13.01 | 13.46 |

Table 4: Using Out-of-Domain Monolingual Data

| experiment | TED (source) | UN (target) |
|---|---|---|
| parallel 10k | 9.46 | - |
| + 10k WoW | 4.13 | 6.16 |
| + 20k WoW | 4.2 | 7.37 |
| + 40k WoW | 4.85 | 7.51 |

Table 5: Using Out-Of-Domain Test Set for *es-en*

## 5.3 EFFECT OF OUT-OF-DOMAIN MONOLINGUAL DATA

There might be scenarios where we want to perform translations for a specific domain where only a small parallel corpus is available, but monolingual data from a different domain is readily available. In this section, we explore such a setting. While the TED Talks corpus consists of spoken language data (in-domain), monolingual data is drawn from the news domain (*out-of-domain*). We randomly sample 10k, 20k and 40k data from the WMT 2017 News task dataset. Table 4 shows that using out-of-domain monolingual corpora also shows significant improvements over all three baselines. For *de-en*, adding 40K *out-of-domain synthetic* samples achieves the same performance as adding 10K *in-domain parallel* samples.

## 5.4 DOMAIN ADAPTATION

Another realistic scenario is one where parallel corpus is available in a source domain (TED Talks), but we care about translation in a target domain (news), where only monolingual data is available. Methods like BT would not perform well here, since the reverse model has only seen data from the source domain. Moreover, the out-of-vocabulary problem is only exacerbated in such a setting, making our approach even more attractive. Our setup here is the same as in section 5.3, except that we use an out-of-domain (10k pairs from the UN *es-en* corpus[2]) test set. From table 5, we observe that WoW trained on the dataset augmented using monolingual data in the target domain shows impressive gains compared to both the parallel baseline (trained only on the source domain) as well as WoW using just the source domain. Adapting to domains like medicine using domain specific bilingual lexicons would be an interesting line of future work.

| experiment | de-en | es-en |
|---|---|---|
| parallel 10k | 9.46 | 9.51 |
| + 20k WoW | 12.66 | 12.34 |
| + 10k WoW + 10k COPY | 12.09 | 13.03 |
| + 10k WoW + 10k BT | 11.46 | 12.02 |

Table 6: Combining Approaches

## 5.5 COMBINING WITH OTHER APPROACHES

It is trivial to combine WoW with other augmentation approaches like COPY and BT. We hypothesized that combination with COPY is most promising, given that Currey et al. (2017) have shown that the benefits are complimentary to back-translation. Specifically, COPY helps the model copy unchanged words like named entities. We explore such combinations by running experiments for WoW + COPY and WoW + BT (table 6). As expected, combination with COPY performs better since the copied target sentences contains words like named entities which are entirely missing from the bilingual dictionaries. Interestingly, 10k WoW + 10k COPY outperforms using 20k WoW for *es-en*, as the model is able to draw upon the complimentary benefits of both methods. On the other hand, combining our method with BT leads to a decrease in performance, which goes on to show that a low quality BT model does not add any complementary benefit.

---

[2]http://www.statmt.org/wmt11/

## 6 CONCLUSION AND FUTURE WORK

We propose a simple yet effective data augmentation technique by utilizing bilingual dictionaries for low resource NMT. In this work, we used ground truth dictionaries. A direct line of future work is to create synthetic samples using induced dictionaries and also incorporating phrase tables.

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
