# OpenReview forum: "INCORPORATING BILINGUAL DICTIONARIES FOR LOW RESOURCE SEMI-SUPERVISED NEURAL MACHINE TRANSLATION"
_ICLR.cc/2019/Workshop/LLD — LLD 2019_

### Official Review · AnonReviewer2 · 2019-04-05
**Simple technique for improving low-resource translation when bilingual dictionaries are given.**

**Rating:** 3
**Confidence:** 3

**Review:**

This paper investigates the idea of using bilingual dictionaries to create synthetic sources for target-side monolingual data in order to improve over NMT models trained with small amounts of parallel data.
This strategy is compared with back-translation and copying the target to the source side and evaluated on TED data for de-en and es-en in a simulated low-resource and a domain adaptation setting. The empirical results show that when little parallel data is available  in addition to bilingual dictionaries, this method can outperform back-translation and copying.

Pros:
- Written clearly
- Reproducible (hyperparameters, data)
- Evaluation shows improvements of the proposed model over baselines, despite the simplicity of the data and the noise in the sources.
- Effect of data sizes are studied.
- Good review of related work.

Cons:
- The low-resource setting is only simulated. It would have been to take a truely low-resource language and evaluate the methods on that (e.g. the other language pairs presented in Qi et al. 2018).
- The requirement of bilingual dictionaries and their coverage and their domain dependence is not discussed. If little parallel data is available, can we simply assume the existence of large dictionaries?
- It is assumed that the word-by-word dictionary translation "at least ensures that the words in the synthetic sentences are accurate" (§4). This is critical since it ignores the problem of polysemy - one word in the target language can often have more than one meaning in the source language: which one is picked for generating the synthetic sentence?

In summary, despite its clarity and simplicity, I don't find the paper very creative regarding the methodology, and it does not sufficiently answer the question when dictionaries outperform back-translation, since the properties of the additional resource, i.e. the dictionary, are not discussed/investigated, neither its limitations. It would have been interesting to see the same approach in a truely low-resource problem where the dictionary might be limited as well.

Details:
- Consider changing the acronym of the method, it seems widely adopted for World of Warcraft.
- Table 1 has encoding problems for á, ò etc.

---

### Official Review · AnonReviewer1 · 2019-04-06
**Good paper. Approach is simple and intuitive, yet effective. The paper lacks a comparison to/discussion with a closely related work. Requires a few formatting fixes**

**Rating:** 3
**Confidence:** 2

**Review:**

Paper summary:
This paper targets machine translation of low-resource languages, where the main problem of current approaches is dealing with Out-Of-Vocabulary words. Given a small parallel corpus of the source and target language, the authors propose a data augmentation technique using dictionaries of the source-target languages. Specifically, given a relatively small parallel corpus of source and target sentences, and given a new set of sentences in the target language (English, in this work) and a dictionary from the target to the source language, the set of sentences are translated word by word using the dictionary the source language (Germen, and Spanish in this work), then the original sentence and the resulting sentences are added to target and source datasets in the parallel corpus, respectively.
The authors compare their proposed methods to two existing techniques: Back translation, COPY which copies OOV words into the sentence without translation. In their various experiments, the results convey the effectiveness of the proposed approach where it achieves an increase in the BLEU score.

Pros.
1-	The paper suits the workshop domain.
2-	The proposed approach is simple, yet it performs on par with the other baseline methods.
3-	The proposed model is evaluated in different scenarios, and the experimental details are provided in the paper.
4-	Generally, the paper is well-written and easy to follow.
5-	The authors discussed how their proposed method has higher coverage on both the target and source languages, in contrast to the COPY method which targets only the target language. I think this is an important contribution and could replace the third contribution.
6-	The authors discussed one of the potential side effects that word-by-word translation can cause, which is the syntax/grammar correctness of the resulting sentence.
Cons.
1-	I believe this paper should include a comparison with Sennrich et al 2015 below or at least a discussion on why it was excluded. This work was proposed to address the same problem that the authors target. In this work, sub-words are used as the tokens for translation in order to address the OOV problem. Specifically, an external dataset is used to get a list of most common subwords instead of full words.
Sennrich, R., Haddow, B., & Birch, A. (2015). Neural machine translation of rare words with subword units. arXiv preprint arXiv:1508.07909.

2-	Using a dictionary for word by word translation is very similar to the idea of using synonyms to mask a writing style. Both these ideas result in not only syntactic issues but semantic ones as well. For example, ‘Kicked the bucket’ which means ‘passed away’ will lose its meaning if translated word by word. The syntactic part was covered properly in Section 4, while the semantic part was partially covered in Section 5.4 where the authors discuss domain adaptation. In my opinion, (and of course as the results show) using COPY with the authors’ method should be elaborated more, both in the discussion and the experiments.

Additional minor (formatting) issues:
-Table 1 is not clear. First, since the study is performed on two languages, the caption should specify this example is on which language. Second, since different approaches target a different part of the corpus (either the source or the target language) I suggest separating them either in two smaller tables or by having an empty row. What I see in this table is an alternation between languages and I find it a bit confusing.
-Tables 2 and 3. Please add ‘using BLUE score’ to the captions. That would be faster to spot compared to looking for it in the text.

---

### Decision · Program_Chairs · 2019-04-16
**Acceptance Decision**

Accept